# Network Analysis of Enhancer–Promoter Interactions Highlights Cell-Type-Specific Mechanisms of Transcriptional Regulation Variation

**DOI:** 10.3390/ijms25189840

**Published:** 2024-09-11

**Authors:** Justin Koesterich, Jiayi Liu, Sarah E. Williams, Nan Yang, Anat Kreimer

**Affiliations:** 1Graduate Programs in Molecular Biosciences, Rutgers The State University of New Jersey, 604 Allison Rd., Piscataway, NJ 08854, USA; jhk148@scarletmail.rutgers.edu (J.K.); jl2791@scarletmail.rutgers.edu (J.L.); 2Department of Biochemistry and Molecular Biology, Rutgers The State University of New Jersey, 604 Allison Road, Piscataway, NJ 08854, USA; 3Center for Advanced Biotechnology and Medicine, Rutgers The State University of New Jersey, 679 Hoes Lane West, Piscataway, NJ 08854, USA; 4Nash Family Department of Neuroscience, Friedman Brain Institute, Icahn School of Medicine at Mount Sinai, New York, NY 10029, USA; sarah.williams@icahn.mssm.edu (S.E.W.); nan.yang1@mssm.edu (N.Y.); 5Institute of Regenerative Medicine, Icahn School of Medicine at Mount Sinai, New York, NY 10029, USA; 6Alper Center for Neurodevelopment and Regeneration, Icahn School of Medicine at Mount Sinai, New York, NY 10029, USA; 7The Graduate School of Biomedical Sciences, Icahn School of Medicine at Mount Sinai, New York, NY 10029, USA

**Keywords:** disease genetics, epigenetics, enhancer–promoter interaction, autism, cell networks

## Abstract

Gene expression is orchestrated by a complex array of gene regulatory elements that govern transcription in a cell-type-specific manner. Though previously studied, the ability to utilize regulatory elements to identify disrupting variants remains largely elusive. To identify important factors within these regions, we generated enhancer–promoter interaction (EPI) networks and investigated the presence of disease-associated variants that fall within these regions. Our study analyzed six neuronal cell types across neural differentiation, allowing us to examine closely related cell types and across differentiation stages. Our results expand upon previous findings of cell-type specificity of enhancer, promoter, and transcription factor binding sites. Notably, we find that regulatory regions within EPI networks can identify the enrichment of variants associated with neuropsychiatric disorders within specific cell types and network sub-structures. This enrichment within sub-structures can allow for a better understanding of potential mechanisms by which variants may disrupt transcription. Together, our findings suggest that EPIs can be leveraged to better understand cell-type-specific regulatory architecture and used as a selection method for disease-associated variants to be tested in future functional assays. Combined with these future functional characterization assays, EPIs can be used to better identify and characterize regulatory variants’ effects on such networks and model their mechanisms of gene regulation disruption across different disorders. Such findings can be applied in practical settings, such as diagnostic tools and drug development.

## 1. Introduction

The vast majority of disease-associated variants are found in non-coding regions of the human genome, as reported in Genome-Wide Association Studies (GWAS) [1]. Significant progress has been made in mapping non-coding genomic regions, such as regulatory regions [2,3]; however, there is still much to uncover. Enhancers and promoters are two regulatory elements that are central to the transcription process. It is known that promoters act as the landing site for RNA polymerase II and as the start site of transcription (TSS) [4], while enhancers are known to harbor transcription factor (TF) binding sites to recruit TFs and increase the efficiency at which transcription occurs [5]. It is then hypothesized that disruptions to either of these elements can lead to inefficient transcription and an imbalance of gene products, contributing to disease risk. However, it is challenging to investigate variant disruption to enhancer and promoter regions due to cell-type-specific transcriptomic variability. This is, in part, due to limitations in identifying active enhancers and promoters, since their activity level and interactions change between cell types and conditions. Additionally, enhancers can be located far away from their target promoter and do not always interact with the nearest promoter region [6]. It thus impedes efforts to properly quantify the disruption that disease-associated variants in these regions are inflicting on transcription. As many neurological disorders often result from multiple small effecting disruptions, it is critical to be able to separate true transcriptional disruption from cell-type variability noise and model their effect.

To this end, our study identifies and characterizes enhancer–promoter interaction (EPI) networks, and how they vary across different cell types. Utilizing the widely used Activity By Contact (ABC) model [7], we incorporate both markers for activity and accessibility, and spatial information to effectively predict EPIs [8,9,10,11,12,13]. Our data consist of six human in vitro differentiated neuronal cell types across four published studies [10,14,15,16], spanning neural developmental stages from embryonic stem cells to neural progenitor cells, and to mature neurons (Figure 1). We selected only neuronal cells to focus our findings on differences between closely related cell types rather than broad differences between cell lineages. Notably, we investigate how neurodevelopmental disorder-associated variants are enriched across these neuronal cell-type-relevant networks.

We first compare the rate of cell-type specificity for each of the elements of these EPIs to identify elements that are cell-type specific, differentiation-stage specific, or ubiquitously found. We next analyze changes in the TFs predicted to bind to our enhancers to identify how TFs influence the cell-type specificity of enhancers. Finally, we investigate the enrichment of potential disease-associated variants within these cell-type EPI networks and their specific sub-structures. By analyzing our EPI networks in this manner, we confirm that our findings regarding the rate of cell-type specificity regulatory elements and their enrichment in relevant biological processes across these cell types are in concordance with previous studies [15,17,18]. 

Our results suggest that enhancers rather than promoters, and TFs that are computationally predicted to bind to enhancers, are the regulatory elements showing higher cell-type specificity. Such TFs, however, are shown to separate between progenitor and mature neuronal cell types, and promoters are still able to identify enrichments for relevant biological processes in each cell type. Interestingly, these EPIs are significantly enriched with implicated disease-associated variants within relevant cell types. Additionally, these variants are enriched in specific sub-structures of EPIs, which suggest possible insights into their disruption mechanism. Our study suggests a novel selection method; as high-throughput data continue to identify additional disease-associated variants. This framework can be utilized for prioritizing variants and their likely cell-type environments to be tested with functional assays [19].

## 2. Results

### 2.1. Generation of Enhancer Promoter Interactions Network

To identify the networks of active Enhancer–Promoter Interactions (EPIs), we first obtained activity and accessibility datasets across six neuronal cell types (Section 4.2). In detail, the accessibility assays encompassed ATAC-Seq for profiling transposase-accessible chromatin [12,13] and Hi-C for mapping chromosomal conformation [20]. The activity assays include H3K27ac ChIP-Seq or Cut&Run for inferring active enhancer regions and RNA-seq for quantifying target gene transcripts [21]. These cell types include three progenitor stages: Embryonic Stem Cells (ESC) [14], Neural Stem Cells (NSC) [14], and Neural Progenitor Cells (NPC) [16] along with three mature neuronal cell types: excitatory neurons (Ngn2) [15], inhibitory neurons (AD) [15], and motor neurons (Motor) [10] (Figure 1). Each cell type was analyzed individually and then categorized into two groups for additional comparative analyses: Progenitor (ESC, NSC, and NPC) and Mature neuron (AD, Ngn2, Motor). 

Next, we utilized the Activity by Contact (ABC) model to quantify enhancer–promoter interactions within a five-mega-base range of each promoter for each cell type. Briefly, the ABC model is a computational approach that integrates accessibility and activity data to identify active enhancer regions, investigate their proximity to active promoters, and generate a network of predicted interacting enhancer and promoter pairs within a cell type. After generating the EPI networks with the ABC model, we excluded non-significant interactions and interactions yielding a low gene product by removing EPIs with an ABC score < 0.02 and a target-gene RNA expression of less than 1 transcript per million (TPM) (Methods; Figure 1). At this stage, we opted not to further filter or categorize the EPIs based on their presence across multiple cell types, thereby preserving the generalizability of our findings to any dataset generated via similar methodologies. Collectively, the union of the six cell-type EPI networks resulted in a *collapsed EPI network* comprising 95,370 EPIs. This network encompasses 51,688 unique enhancer regions (30,767 when merging overlapping enhancers) and 11,578 unique promoter regions, offering a substantial foundation for exploring cellular differentiation and cell-type-specific regulatory mechanisms (Figure 2A; Appendix A). Importantly, we employed several approaches to control for potential batch effects between these studies (Methods).

Moving forward, our analysis focuses on three levels of comparison. We start with generating the ABC-predicted EPI networks separately for each of the six cell types across neural differentiation and the creation of the *collapsed EPI network*. Our primary comparison is the ABC-predicted EPI output of the six cell types to each other. This analysis includes duplicate EPIs predicted across multiple cell types and focuses on identifying cell-type-specific factors without including any cell-type comparison filters. This allows each cell type’s findings to be independent, to investigate if EPIs can characterize individual cell types on their own, without the need to compare them to another cell type. Our second level of comparison utilizes the same EPI network but in addition to analyzing the six cell types separately, we group the findings of the three undifferentiated cell-type EPIs and the three mature neuron EPIs to investigate progenitor versus mature neuron enrichments. This analysis will help distinguish factors of the EPIs acting in a cell-type-specific manner, a differentiation-stage manner, and a ubiquitously expressed manner. Our third comparison takes the *collapsed EPI network* and subsets the EPIs based on their cell-type or differentiation-stage specific regulatory elements (Methods). While this approach focuses on cell-type-specific elements, it requires additional cell types to filter the EPIs.

### 2.2. Experimental Data Validate EPIs Cell-Type Specificity

First, we aimed to determine the corroboration of our predicted cell-type-specific EPIs through additional experimental studies. To this end, we examined the expression level of cell-type-specific marker genes and their presence in our predicted networks. Specifically, we used gene expression data from Inoue and Kreimer et al. 2019 [22], which included differential expression analysis between ESCs and ESC-derived NPCs (Methods). We examined the enrichment of genes that are differentially expressed between these ESCs and NPCs (Methods) and their presence in our ESC and NPC EPI networks. Our results indicate that genes highly expressed in ESCs are enriched in our ESC EPI network compared to our NPC EPI network (Fisher’s exact test *p*-value: 1.444 × 10^−4^, Odds Ratio: 1.15). Similarly, genes highly expressed in ESC-derived NPCs are enriched in our NPC EPI network compared to the ESC EPI network (Fisher’s exact test *p*-value: 1.056 × 10^−45^, Odds Ratio: 1.86). These findings support the cell-type specificity of our EPI networks.

### 2.3. Cell-Type Specificity of EPIs Is Driven More by Enhancers than Promoters

Since we find a large proportion of active enhancers and promoters in multiple cell types (Figure 2B; Appendix A), we set out to identify the contribution of enhancers and promoters to the cell-type-specificity of EPIs. Comparing the percentage of cell-type-specific enhancers and promoters, we found an average of 35% of enhancers that are unique to each cell type. On the contrary, we found that promoters exhibit less than 5% uniqueness to an individual cell type on average (Figure 2B; Methods). Furthermore, we examined the proportion of overlapped regulatory elements between cell types utilizing the Jaccard index (Figure 2C; Methods) [23]. For both the enhancers and promoters, we observe that the highest overlap of regulatory elements between cell types is within the differentiation stages. Excluding this, we identify that enhancers, compared to promoters, have a notably lower amount of overlap between any two cell types (Figure 2C, Appendix A). We note that our analysis finds instances where cell types across published studies relate more closely than results of cell types from the same study. These comparisons highlight that our findings are biological differences rather than technical differences.

To gain a deeper understanding of enhancers and promoters’ cell-type specificity, we analyzed a subset dataset of the EPIs in parallel to the above-mentioned EPI network. This particular subset, referred to as the *subset network*, delineates the initially constructed EPIs into nine distinct categories, with a particular emphasis on separating cell-type-specific EPIs (Methods). Therefore, these nine categories of EPIs encapsulate unique identifiers for each cell type, stages of progenitor development, mature neurons, and likely common ubiquitous housekeeping EPIs. Within this subset network analysis, we observed an increased number of unique enhancers within each of the nine cell-type categories (Appendix A) compared to the findings of the collapsed network EPI categories (Figure 2B), while the proportion of unique promoters remains largely unchanged (Appendix A). This is consistent with the fact that the subset network removes duplicate EPIs found in multiple cell types that would increase the number of overlapping regions across cell types in the collapsed EPI network analysis.

Additionally, similar to our previous observation regarding cell-type EPIs, the greatest extent of overlap for both enhancers and promoters occurred within the differentiation stages. When excluding these instances, enhancers demonstrated a significantly low overlap across cell types, in contrast to promoters, which maintained a higher level of commonality between any two cell types (Appendix A). These comprehensive analyses, applied to both the overarching and subset-specific EPI networks, yielded congruent outcomes, highlighting the nuanced dynamics of enhancers driving a subset of target genes to activate cell differentiation stage and cell-type-specific processes. 

Finally, to provide further biological insight into our predicted EPIs into cell types and differentiation stages, we analyzed the target genes associated with promoters identified within the aggregated EPI networks. By employing gene set enrichment analyses, we identified the top five enriched Gene Ontology (GO) biological processes for each cell type (Figure 3A) [24]. Our findings indicate a predominant enrichment of processes associated with neuronal functions. Specifically, we observed an enrichment for protein localization and proteasomal activity in Embryonic Stem Cells (ESCs), and for developmental Wnt signaling in Neural Stem Cells (NSCs), a pathway integral to the early stages of cell differentiation and absent in mature neurons [25]. Furthermore, we examined the comparative influence of cell-type-specific enhancers and promoters on these observed enrichments. The only significant enrichment for the cell-type-specific promoters was identified as non-cell-type-relevant mesenchymal morphogenesis processes in motor neurons. Promoters targeted by cell-type-specific enhancers, however, largely replicate the biological processes identified in the cell-type-collapsed EPI gene set (Figure 3B). These results underscore the pivotal role of cell-type-specific enhancers in the regulation of genes that are enriched for cell-type-specific processes, highlighting their critical function in the orchestration of cell-type-specific transcriptional regulation. 

### 2.4. Transcription Factor Binding Presence Distinguishes between Differentiation Stages 

After identifying the importance of enhancers in cell-type specificity, we further explored the transcription factor binding sites (TFBS) within our predicted enhancer regions, since enhancers mainly act as binding hubs for TFs [26,27]. In detail, we scanned for TFBS within the ABC-model-predicted enhancer regions by utilizing the Find Individual Motif Occurrences (FIMO) [28] tool (Methods). Using a binding site prediction confidence *p*-value of less than 10^−5^, we revealed a total of 3,515,990 TFBS across the majority of enhancer regions (only 3 enhancers did not have a predicted binding site), consisting of 602 unique TFs, underscoring enhancers’ role as central nodes in the transcriptional regulatory network. Despite the extensive prediction of TFBS, no cell-type-specific TFBS patterns emerged, with the TF presence distributed broadly across enhancers and cell types. To delve deeper into the role of TFs in dictating cell-type-specific regulatory functions, we analyzed TF gene expression patterns across the six cell types (Methods). While TF gene expression clustered by cell type (Appendix A), they did not correspond to cell-type-specific enriched processes (Appendix A). However, differentiation stage-specific clustering revealed unique enrichments: TFs associated with progenitor cells are significantly enriched for only neuron and brain development, whereas TFs in mature neuron cell types showed enrichment for gland and skeletal system development (Appendix A).

An additional method to cluster the TFs utilized the presence of TFBSs in enhancers to identify TFs with a variable presence across cell types (Methods). While we were unable to identify any significant enrichment of TFs based on their raw binding presence (Appendix A), we were able to identify significant enrichment using their weighted binding presence (Methods). We find that TFs have a similar amount of presence within the same cell types of a differentiation stage but vary across differentiation stages (Figure 4A). We thus clustered the TFs based on differentiation stages rather than individual cell types (Methods). We identify TFs clustered with higher ranked presence in progenitor cells to have unique enrichments for in utero embryonic development, signaling processes, and primary microRNA transcription, which has previously been implicated to be active in early development cell types (Figure 4B) [29]. In accordance with the prior TF gene expression clustering analysis (Appendix A), TFs clustered for higher-ranked presence in mature neurons have enrichment for skeletal system development (Figure 4B). These findings collectively suggest that the predicted binding of TFs cannot indicate cell-type-specific processes, but their gene expression and weighted presence in enhancers can indicate differentiation stage-specific biological processes [9]. 

### 2.5. Characterization of Enhancer–Promoter Interactions Architecture Reveals Cell-Type-Specific Regulatory Mechanisms

To better understand how the enhancers might select their target promoters to create cell-type-relevant enrichments, we investigated the patterns of enhancer–promoter interactions across the six neuronal cell types. First, we define four categories of EPI sub-structures: (1) a single enhancer regulating one promoter (labeled as “C1”); (2) a single enhancer regulating multiple promoters (labeled as “C2”); (3) multiple enhancers regulating a single promoter (C3), and (4) multiple enhancers regulating multiple promoters (C4) (Figure 5A; Methods). Second, we scrutinized the percentage of each sub-structure category in each cell type. Our results showed a similar proportion of EPI sub-structures across different cell types, with the majority, at around 53%, consisting of category C4: multiple enhancers regulating multiple promoters. The next largest cluster was category C3 at ~26%, followed by category C2 at 16%, and the smallest, which was the single enhancer to the single promoter, C1 at 5% (Figure 5B). However, with the subset EPI network, we note that there is a more equal distribution of the interactions among the four sub-structures than seen in the collapsed EPI network (Figure 5C). For the subset EPIs, we find the largest section is now C3 at ~30%, C4 at 26.5%, C2 at 25%, and C1 at 18.5%. With the subset EPI network, we see significant shifts from the multiple enhancers regulating a multiple promoter sub-structure (label C4) toward the single enhancer regulating single or multiple promoters (labels C1 and C2). This shift suggests that the aggregated EPI network integrates cell-type-specific enhancers with those active during different differentiation stages or expressed ubiquitously [30]. Sub-setting EPIs allows for distinguishing cell-type-specific enhancer interactions from those more broadly active, resulting in a more nuanced understanding of enhancer–promoter dynamics. 

### 2.6. EPIs Can Be Utilized to Prioritize Potential Disease-Associated Variants

Given the potential of enhancer–promoter interactions (EPIs) to delineate cell-type-specific regulatory mechanisms relevant to biological processes, we explored the utility of EPIs in identifying and prioritizing genetic variants associated with disease. Particularly, we aim to determine whether disease-associated variants are disproportionately represented within EPIs, thereby providing insights into their potential to influence disease phenotypes. Utilizing an empirical *p*-value test, we assessed the enrichment of disease-associated variants within active enhancer and promoter regions, and downstream target genes of EPIs, relative to a random set of all predicted EPIs (Methods). 

Specifically, our analysis included a dataset of approximately 250,000 autism spectrum disorder (ASD) de novo variants from both ASD patients (cases) and healthy siblings (controls) [31]. We found significant enrichment of enhancer regions containing ASD de novo variants in neural progenitor cell (NPC) and astrocyte-derived (AD) cell types, with nominal significance in neural stem cells (NSC) (Figure 6A; Figure 7A; Appendix A). Notably, these enhancer regions are not enriched for control de novo variants, suggesting a higher likelihood of disruption in these specific cell types (Figure 6A; Figure 7C; Appendix A).

On the contrary, no cell types showed enrichment for promoter regions overlapping with either case or control variants, aligning with our previous observations of variant enrichment within promoter regions but not corroborated by functional assays (Figure 6B; Figure 7B,D; Appendix A) [32]. Moreover, we assessed the enrichment of EPI target genes against a catalog of known ASD-associated genes, identifying significant associations in the NPC, AD, Ngn2, and Motor EPI networks with ASD-related genes (Figure 6C; Figure 7G; Appendix A). Further comparative analysis of EPIs for case versus control variants across the EPI sub-structure of the six cell types revealed distinct patterns of variant enrichment in enhancer and promoter regions, suggesting differential regulatory roles in disease manifestation. For control variants, we found significant enrichment for the category C4 EPIs (multiple enhancers regulating multiple promoters) (Figure 5A; label C4) (Fisher exact test *p*-value: 0.00276, Odds ratio: 1.116). However, for ASD case variants, category C3 EPIs are significantly enriched (multiple enhancers regulating a single promoter) (Figure 5A, label C3) (Fisher’s exact test *p*-value: 0.00396, Odds ratio: 1.091). In addition, we also identified increased ASD case variants within the C2 category, where a single enhancer regulates multiple promoters (Figure 5A, label C2) (Fisher’s exact test *p*-value: 0.0309, Odds ratio: 1.165), with a nominal significance *p*-value. In summary, variants in C2 and C3 exhibit significant case enrichments while C4 displays significant control variant enrichment. 

Expanding our analysis to include schizophrenia-associated variants (SCZ) and neurodevelopmental disorder (NDD) genes ([33,34] Methods), we observed nominally significant enrichment of schizophrenia-associated variants among enhancer regions of EPIs in the inhibitory neuron cell-type AD and promoter regions of ESC cell-type EPIs (Figure 6A,B; Figure 7E,F; Appendix A). These findings are in line with previous studies highlighting schizophrenia as affecting neuron development as well as inhibitory neurons [35,36]. Additionally, when analyzing the cell types for the enrichment of NDDs, we found all six cell types significantly enriched (Figure 6C; Figure 7H; Appendix A). This finding is also supported by Fu et al. [33], who identify the enrichment of NDD genes in early developmental neuronal cells, while the ASD genes also included in the NDD gene list had more enrichment in the later development and mature neurons [33].

In conclusion, these findings highlight the potential of our enhancer–promoter interaction analysis in the prioritization of disease-associated variants, offering a novel avenue for understanding the regulatory underpinnings of complex disorders.

## 3. Discussion

We set out to investigate the nature of EPIs and how they can indicate cell-type-specific mechanisms of disease-associated variants on transcriptional regulation. To this end, we used six neural cell types spanning different neuronal differentiation stages. We first show that EPIs are acting in a cell-type-specific manner. We identified that, in accordance with previous studies, a significant percentage, ~30%, of the predicted EPIs include enhancers specific to one cell type, whereas promoters are mostly shared across cell types. Interestingly, we find that promotors can identify cell-type-specific biological processes. However, these enrichments are not determined by cell-type-specific promoters but rather by cell-type-specific enhancers regulating a particular set of promoters. These results highlight the complex combinatorial nature of gene regulation mechanisms. 

In addition to identifying enhancers as an important determinant of enrichment for cell-type-relevant biological processes, we found that the presence of predicted TFBSs does not drive this finding. Instead, our analysis suggests that the weighted presence of TFBSs in enhancers and TF gene expression are better indicators for differentiation stage processes than cell-type-specific processes. This highlights that the process by which enhancers regulate transcription is complex and requires additional research for a better understanding of all the factors involved.

We next set out to identify the structures in which enhancers interact with promoter regions. When separating EPIs by cell type, we identified an increase in a single enhancer regulating a single promoter structure while also observing a decrease in the multiple enhancers regulating single/multiple promoter structures. This leads us to hypothesize that some promoters that are active in multiple cell types are regulated by a combination of cell-type-specific along with differentiation-stage-specific enhancers, highlighting the need for future research on enhancer activity to be conducted in a cell-type-specific resolution.

Finally, we investigated if these EPIs can be utilized as a selection method for future functional assays of potential disease-associated variants. We show that each part of the predicted EPIs: enhancers, promoter regions, and target genes, can be utilized as a selection mechanism for disease-associated variants and genes with potentially increased likelihood of disease effects in a given cell type. We find a significantly higher number of enhancers with a sub-structure of multiple enhancers to multiple promoters enriched with ASD control variants. Conversely, promoter regions that belong to the multiple enhancers to a single-promoter sub-structure are enriched with ASD case variants. Additionally, we also identified enhancers in the sub-structure of a single enhancer regulating multiple promoters to be enriched with ASD case variants. Together, these findings suggest the possibility of compensatory mechanisms that prevent these control variants located in enhancers from causing disease-related disruptions. While in the promoter and enhancer regions, where there is no other promoter or enhancer in the sub-structure to compensate for the disruption, there is a potential for the variant to lead to disease-related dysfunction in these cells. While we cannot definitively say whether this proposed selection method identifies higher disruptive variants, we posit that this is a promising starting point and will be the focus of future analyses when more functional data are available on these variants within the reported cell types.

We note that this study has a few limitations. The first being that the EPI prediction is performed computationally and future assays confirming these interactions will improve our confidence in these findings. Second, the EPI prediction requires, at a minimum, cell-type-specific ATAC-seq and H3K27ac ChIP-seq, which are not available for all cell types. Third, future analysis of the functional characterization of disease-associated variants would be required to test the hypothesis that EPI-overlapping variants are likely to be more disruptive than non-overlapping ones. Finally, it would also benefit this analysis to investigate the variability of these EPI interactions within the cell type between healthy and disease cell environments. However, we currently do not have epigenetic data of all the tested cell types from disorder-afflicted patients.

With these findings, we gain a deeper understanding as to what factors lead to cell-type-specific transcription, the type of interactions in which these structures occur, and that these EPIs can help differentiate cell types that are more likely to show disruptions caused by potential disease-associated variants. These findings provide more insight into the complexity of transcription variability [8,9,10,11] and as high-throughput data continue to identify increasing numbers of potential disease-associated regulatory variants, these findings will help improve the selection and prioritization of these variants for testing using functional assays [19]. 

## 4. Methods

### 4.1. Review of Terminologies Used in This Study

The key concepts and terminologies related to transcriptional regulation are crucial for this study. This Section provides explanations of the major terms used throughout the manuscript to facilitate a clear understanding of the research findings. 

Activity-By-Contact model (ABC model): The principle of the ABC model is that an enhancer, which is accessible, highly active, and has a high contact frequency with a promoter, is also likely to have a regulatory impact on that promoter’s gene. Activity-By-Contact score (ABC score) is utilized to describe the activity of the EPI. To calculate the scores, the necessary inputs are to replicate separated ATAC-Seq peak files, replicate merged H3K27ac peak files, and the RNA-Seq-processed TPM table file (described in the following Section). The formula of ABC score is the activity of the enhancer, defined by the read counts of the ATAC-seq and H3K27ac peaks, multiplied by the contact frequency between the enhancer and target-gene promoter, defined by the Hi-C contact frequency or estimated distance frequency, divided by the sum of all enhancer scores between the target gene promoter and enhancers within 5 mega-bases of the target gene promoter. The Enhancer–Promoter Interaction, abbreviated as EPI, is defined as “interactions between *cis*-acting enhancer and promoter that facilitate enhancer-mediated upregulation of gene transcription”. In our study, an active EPI is confirmed when its Activity-By-Contact score (described below) is greater than the threshold 0.02 and the enhancer is within 5 mega-bases of the target-gene promoter. Collapsed EPI network: To investigate EPIs identified in different cell types, we concatenated the 6 cell-type EPI networks into a single collapsed EPI network without additional filtering to investigate differences between cell-type EPIs and as a pooled resource to select from for the empirical analysis of overlapping potential disease-associated variants and genes. Further details regarding the generation of the EPI network and the empirical analysis are described below.Subset EPI: To study more cell-type-specific EPIs, we subset the EPIs that were initially identified in this study. Briefly, the criteria for sub-setting are based on ABC scores in specific combinations of cell types such as only one cell type, at least two progenitors and no mature neurons or two mature neurons and no progenitor cell types, or ubiquitously across most cell types. Details regarding sub-setting are described in the following Sections.

### 4.2. Dataset Overview 

We utilized publicly available datasets that include the ATAC-Seq, RNA-seq, and H3K27c ChIP-seq or CUT&RUN data of the six studied cell types (Table 1): 

### 4.3. Processing of ATAC-Seq

To remove potential adapter and trailing sequence contamination in the ATAC-seq raw data, we trimmed the reads using trimmomatic (v0.38) [37] with the following parameters: Adapter TruSeq3 with seed mismatches set to 2, palindrome clip threshold of 15, simple clip threshold of 4, minimum adapter length equal to 4, and keep both reads equal to true. Adapter NEBNextPrimers1and2 with seed mismatches set to 2, palindrome clip threshold of 30, and a simple clip threshold of 10. LEADING parameter set to 20. TRAILING parameter equal to 20. SLIDINGWINDOW set to 4:15. And MINLEN parameter set to 25. We next utilized the FASTQC pipeline (Version 0.3.2; Babraham Bioinformatics) on the reads and aligned them to the reference genome (hg19) with bowtie version 2.4.1 [38]. For the bowtie command, we included the following parameters: limit the alignment length to 2000 bp [−X 2000], and remove unpaired [--no-mixed] and discordant alignments [--no-discordant]. We then removed duplicate data and reads in blacklisted regions [39] with GATK [40]. Finally, we call the peak regions with the MACS2 command [21] and use a lenient threshold of 0.1 as well as the regions that appear in all replicates to retain only confidently called peaks. To normalize the peaks called across all cell types and to follow the recommendations of the ABC model, we normalized all ATAC-seq peaks to 500 bp centered on its apex. We then utilized these consolidated and normalized ATAC-seq peaks for the ABC model prediction of enhancer–promoter interactions.

### 4.4. Processing of ChIP-Seq and CUT&RUN Data

The processing for the H3K27ac ChIP-seq or CUT&RUN [41] peak data was performed in a similar fashion as the ATAC-seq peaks with a few notable distinctions. Firstly, to adjust for the size difference in the input read size from the assay, the bowtie command had altered parameters for CUT&RUN. These parameter changes included limiting alignment length to 700 bp [−X 700] and a minimum valid insert size for paired-end alignments set to 10 bp [–I 10]. Additionally, for CUT&RUN data, when running MACS2, we included the parameters that chose not to build a shifting model [--nomodel] and we kept all duplicate reads [--keep-dup all]. After retaining only the regions of MACS2 peaks that had a *p*-value below 0.1, we did not consolidate across replicates as the ABC model required the ChIP-seq or CUT&RUN input to have one file for each replicate. We also did not normalize the peaks to 500 bp surrounding the central point but retained the full peak length. This was performed as the ATAC-seq required higher precision of accessibility region, which was not required for the ChIP-seq or CUT&RUN data. 

### 4.5. Description of Cell-Type-Specific Hi-C Data

Due to the unavailability of Hi-C data for the ESC, NSC, AD, and Ngn2 cell types, we employed a validated power law estimate of contact with respect to genomic distance to ensure analytical consistency, which is the default option when no Hi-C data are provided to the ABC model command [7]. 

### 4.6. Description of Cell-Type-Specific RNA-Seq Data

RNA-seq data were collected from the GEO accession dataset. The files collected were in a 2-column format with the first column describing the gene HGNC symbol identifiers and the second column having the RNA-seq normalized count in TPM form.

### 4.7. Predicting Enhancer–Promoter Interactions Using the Activity-by-Contact Model

We utilized the ABC model to predict enhancer–gene connections in each cell type, based on measurements of chromatin accessibility (ATAC-seq), histone modifications (H3K27ac; CUT&RUN), and chromatin conformation (Hi-C) as previously described. For the ABC pipeline, we called the run.neighborhoods command, supplying it with the consolidated and normalized ATAC-seq peaks, the ChIP-seq or CUT&RUN replicate separated MACS2 filtered output files, the collected RNA-seq TPM table, and the path to the TSS location bed file (hg19 version) that is downloaded along with the ABC command (refGene, version 2017-03-08). By not supplying any Hi-C data, the program defaulted to their validated ten-cell-averaged Hi-C dataset. We then ran the predict command with a default-recommended score threshold of 0.02 to generate our EPIs for each cell type. We then filtered the EPIs further by only retaining EPIs in which the promoter’s downstream target gene had a TPM value greater than 1 in the respective cell type. The 0.02 EPI interaction score threshold is the value recommended in the ABC model publication, which removes interactions that are very weak and unlikely to create any meaningful transcription complexes [7]. Additionally, we apply the TPM filtering step to remove any interactions that lead to a very low amount of gene product. Disruptions to these EPIs, and thus the amount of downstream gene product, would likely not lead to a significant disruption in the cell environment.

### 4.8. Generation of the Collapsed EPI Network

We collected the EPI networks of the 6 cell types from the ABC output file “EnhancerPredictionsFull.txt”. We did minimal filtering by removing EPIs that had a reported ABC score below 0.02 or if the target gene of the EPI had a gene expression of less than 1 transcript per million in the cell type. We then concatenated the EPIs from the 6 cell types into a collapsed network that we utilized in our analysis (Appendix A). For the analyses comparing enhancers across cell types, we took the enhancer regions and read them into R. We then converted the data into a GenomicRanges object and appended all of the cell types together into a single object with labels identifying the original cell type from which the EPI originated. Finally, the enhancers were collapsed using the GenomicRanges command reduce() to merge enhancer regions that overlap, and this enhancer list was used in our analyses. 

### 4.9. Generation of Subset EPI Network

For our supplementary analysis in which we subset the EPIs, we follow similar procedures as our main EPI network generation with minor changes. For each cell type, we take the “EnhancerPredictionsFull.txt” output file from the ABC predict command and read the file into R. We then converted the data into a GenomicRanges object and appended all of the cell types together into a single object with labels identifying the original cell type of the EPI and the ABC interaction score. We then collapse the EPIs using the GenomicRanges command reduce() to merge EPIs with enhancer regions within 100 bp of each other. We then collect the different cell-type ABC scores for the EPIs using the findOverlaps() command between the collapsed and original EPI lists and output the resulting data. We then subset the EPIs as such: 6 subsets where the EPIs only had a non-zero ABC interaction score within one cell type; 1 cluster where the EPIs had a non-zero ABC interaction score in at least 2 of the 3 progenitor cell types (ESC, NSC, and NPC); 1 cluster where the EPIs had a non-zero ABC interaction score in at least 2 of the 3 mature neurons (AD, Ngn2, and Motor); and 1 cluster where the EPIs had a non-zero ABC interaction score in at least 5 of the 6 cell types, herein referred to as the ubiquitous cluster (Ubi). Any EPIs that had a non-zero ABC interaction score in 4 or fewer cell types but in at least 1 progenitor and 1 mature neuron cell type were removed from further analysis.

### 4.10. Fisher’s Exact Test of Enriched Overlap of Cell-Type-Predicted Active EPI with Log-Fold Change

DESeq2 Log-Fold Change (LFC) data comparing ESC RNA-seq and ESC-derived NPC RNA-seq was obtained from Inoue and Kreimer et. al. 2019 [22]. Briefly, read counts for both time points were read into DESeq2 and generated log-fold-change (LFC) and false-discovery-rate (FDR) values for genes in the pairwise comparison. Genes were then filtered to an FDR score < 0.05 and no filter was placed on the LFC score.

The LFC was structured such that higher gene expression in the NPC cell type compared to the ESC cell type had a positive value, while a higher gene expression in the ESC than the NPC cell type generated a negative LFC value. For both analyses we compared the number of genes with significant LFC for that cell type, LFC > 1 for NPC and LFC < −1 for ESC, to all other genes tested. In the 2 × 2 table that makes up Fisher’s exact test, the columns were LFC > 1, LFC < 1, and LFC < −1, LFC > −1 for NPC and ESC active analyses, respectively. For the rows in Fisher’s test analysis, the first comparison was the number of genes found in EPIs predicted to be active in ESC vs. EPIs predicted to be active in NPCs. The second comparison was the proportion of genes predicted to be in active EPIs within the cell type vs. EPIs predicted to not be active in that cell type.

### 4.11. Controlling for Batch Effects across Studies

To control for potential batch effects between the published studies affecting our inter-cell-type analysis, we took the following measures. To remove potential batch effects in the processing or aligning of the assays, all data collected for the cell types were in their raw FASTA format and processed using the same steps. Additionally, we attempted to prevent any batch effect relating to differences in the strength or read depth of certain regions in the assays across the cell types. This was performed by generating each cell-type EPI network individually, and then the cell-type EPI on the binary classification of the presence or absence of the EPI across cell types was assessed, rather than comparing the strength of the ABC interaction score. 

### 4.12. Identification of Enhancers and Genes Predicted Active across Different Cell Types

Enhancers and Promoters for each cell type were predicted as mentioned above. The Bedtools [42] command—intersect—was used to identify if an enhancer had at least 1 base pair overlap with an enhancer in another cell type. For promoters, the grep command was utilized to identify if the HGNC gene names were predicted to be active in other cell types.

### 4.13. Gene Ontology (GO) Enrichment

We tested GO enrichment [24] on EPI target genes, and transcription factors predicted to bind to the enhancers. We first read the HGNC gene symbols into R, then utilized the “org.Hs.eg.db” library to convert the HGNC symbols to Entrez IDs. From there, we utilized the clusterProfiler library to find GO enrichments using the command “enrichGO” with the parameters ont = “BP”, pAdjustMethod = “fdr”, pvalueCutoff = 0.01, qvalueCutoff = 0.05, pool = FALSE

### 4.14. Identifying Sub-Structures of EPIs

To analyze the type of interactions with which the enhancers and promoters interacted, we split the EPIs into four sub-structures. Since sub-structures one and two relied on a single enhancer interacting with promoters, and sub-structures three and four relied on a single promoter interacting with enhancers, we labeled the sub-structures per EPI rather than per enhancer or per promoter. We do note, however, that when we labeled the sub-structures by the enhancer and collapsed all of the types of promoters with which they interacted, we found similar trends to the findings that we reported in the results.

### 4.15. Prediction of TFBSs Using FIMO

Transcription factors predicted to bind to the enhancers and be involved in the EPIs were found using the Find Individual Motif Occurrences (FIMO). We supplied the program with FASTA sequences of the enhancer regions created from the hg19 human genome and the enhancer genomic coordinates. Using the ENCODE motif.meme file, FIMO predicted TF binding sites in our enhancers, and we filtered the bindings to a *p*-value less than 1.0 × 10^−5^. We then collapsed the predicted binding motifs to the TFs that bind to that site, reducing the number of predicted binding sites from 3,170,484 to 378,179.

### 4.16. Clustering TFs Based on Gene Expression

TF gene expression was calculated as the RNA TPM values for the corresponding gene in each of the 6 cell types. TFs were then grouped by the cell type that contained the highest gene expression for the TF gene. Additionally, TFs in the ESC, NSC, and NPC cell-type groups were additionally grouped together in a Progenitor group, while the TFs in the AD, Ngn2, and Motor cell-type groups were grouped into a Mature group.

### 4.17. Identifying the Presence of TFs in Different Enhancer Regions

A binary presence matrix was created to identify TFs present among different enhancers within the different cell types. If the TF was predicted to bind to the enhancer via FIMO, it was represented as a 1 in the matrix, if not predicted to bind, then it was represented as zero. To generate the weighted matrix, we divided the number of enhancers within each cell type to which the TF is predicted to bind by the total number of enhancers that the TF binds to in all 6 cell types. We then visualized the weighted matrix using the Morpheus program [43] and utilized the color gradient and k-mean clustering to separate the TFs into 3 clusters. The 3 clusters are 1. Higher-ranked presence in the progenitor cell types and a lower-ranked presence in the mature cell types. 2. Higher ranked presence in the mature cell types and a lower ranked presence in the progenitor cell types. 3. Approximately equal ranked presence in all cell types.

### 4.18. Selection of Disease-Associated Variants for Empirical Analysis

We collected the hg38 genomic coordinates of the 1/4 million ASD de novo variants as well as hg19 GWAS variants associated with Schizophrenia [31,34]. For the GWAS variants, we also collected all known variants with a linkage disequilibrium greater than 0.80 to any of the GWAS-associated variants. We then used LiftOver to convert the hg19 coordinates to hg38. For the ASD- and NDD-associated gene sets, the gene lists were provided by the authors as simplified gene lists aggregated from the gene columns of their supplementary tables.

### 4.19. Generation of Regulatory Element Regions for Empirical Analysis

We compared the overlap of the enhancers and promoters to disease-associated variants using bedtools intersect, then collapsed each overlap to look at the number of regulatory elements that have at least one variant overlapping. We determined promoter regions as the 2000 bp upstream of the TSS. We collected hg19 genomic coordinates for the target promoters of the EPIs from the RefSeq curated file used to generate the ABC predictions (refGene, version 2017-03-08). For each gene, we determined the promoter region as between 2000 bp upstream of the start coordinate and the start coordinate for genes on the “+” strand on DNA and between the end coordinate of the TSS region and 2000 bp downstream of the end coordinate for genes on the “-“ strand of the DNA. For the overlap between EPI target genes and ASD- or NDD-associated genes, we first generated a unique list of target genes and then collected the number of those gene names that were found in the gene list file using the grep command with the “-x” parameter.

### 4.20. Empirical Analysis of Overlap

For the empirical analysis, we concatenated the ABC prediction output for each cell type after filtering the ABC score for values greater than 0.02 and target genes with a cell-type TPM score greater than 1. We then obtained the unique number of the enhancer regions, promoter regions, and target genes for each cell type. Then, separately for enhancers, promoters, and genes, we randomly selected the same number of enhancer regions, promoter regions, and target genes from the six-cell-type-concatenated-collapsed network and identified the number of regions that contained a variant or associated gene. We calculated the *p*-value as the probability that a random selection of the same number of regulatory regions from any cell type would have equal or more overlaps with the variants than the observed amount for the specific cell type. After 1000 repetitions of randomly selecting regulatory regions and observing the number of overlapping variants, the *p*-value was calculated as 1 (number of instances where the randomly selected group had equal or higher number of overlaps compared to the specific cell-type group/1000). We then used 6 times a multiple testing corrected *p*-value threshold of 0.008333 to determine if the finding is significant and a *p*-value threshold of 0.05 to determine nominal significance.

### 4.21. Code Availability Statement

Scripts utilized in this study for the generation and analysis of EPIs can be found at https://github.com/JKoesterich/Cell_Spec_EPI.

## Figures and Tables

**Figure 1 ijms-25-09840-f001:**
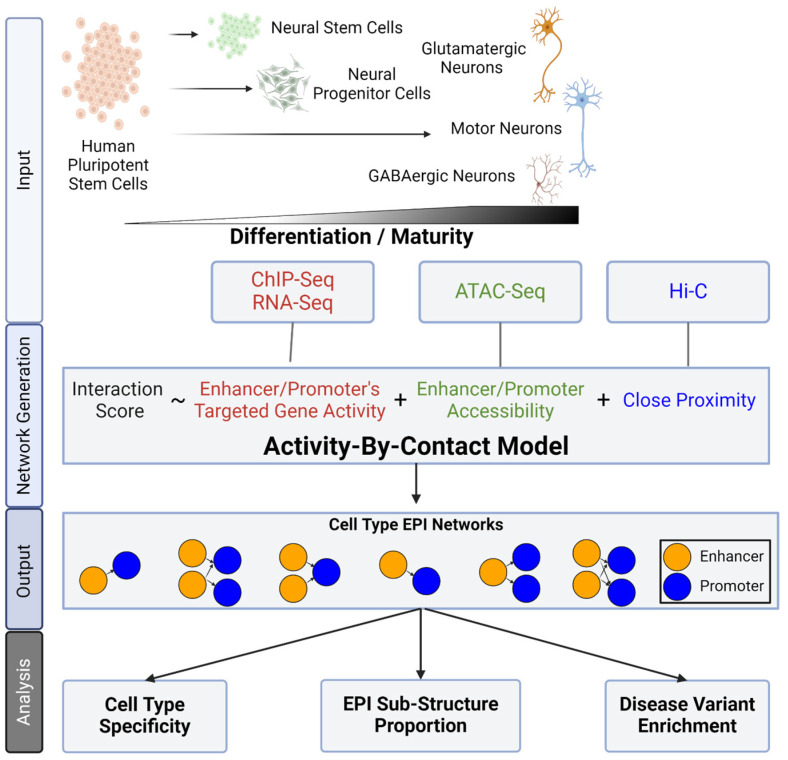
EPI calling: Schematic of cell-type relation, data processing, and analysis steps. The depiction of cell-type relation to the other cell types analyzed in the study, in order of increasing differentiation stage. From left to right: Pluripotent Stem Cells (ESC), Neural Stem Cells (NSC), Neural Progenitor Cells (NPC), Mature Motor neurons (Motor), GABAergic neurons (AD), and glutamatergic neurons (Ngn2). Note that the Motor, AD, and Ngn2 cell types are shown together as they are all different end stages of neuron cell differentiation. The data processing describes the ATAC-Seq, RNA-Seq, and ChIP-Seq or CUT&RUN information collected for each cell type. Due to only having Hi-C data available for the NPC and Motor neurons, we elected to incorporate the ABC model’s validated set of multi-cell-type-averaged Hi-C data [7] for all cell types to keep uniformity of the input data. The analysis steps then describe utilizing the ABC model to generate the cell-type EPI networks, and the types of analysis of such networks.

**Figure 2 ijms-25-09840-f002:**
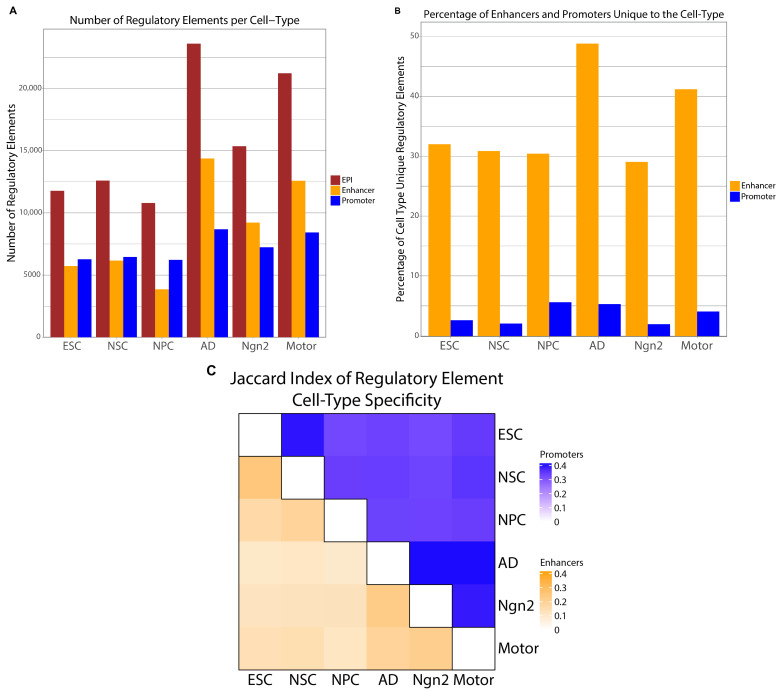
Regulatory elements distribution: (**A**) Grouped bar chart showing the number of total EPIs, total unique enhancers, and total unique promoters per cell type of the collapsed EPI network. (**B**) Bar charts display the percentage of enhancer and promoter regions that are only found within that cell type. (**C**) A heatmap containing Jaccard index values of shared enhancers (bottom left) and promoters (upper right) between any two cell types. A color gradient is created for each regulatory element between 0 and the max Jaccard index value across both the enhancers and promoters. The darker cells represent higher levels of shared elements between the two cell types.

**Figure 3 ijms-25-09840-f003:**
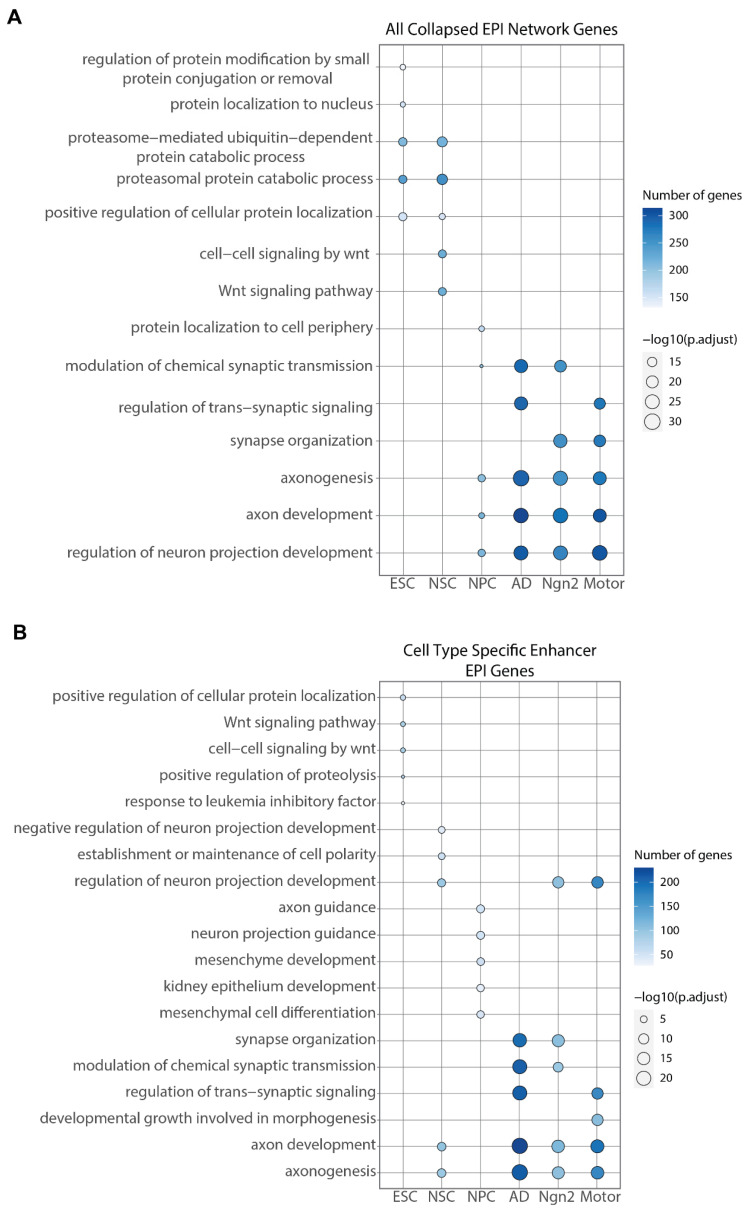
Dotplot figures showing the top 5 enriched biological processes for each cell type. Enrichment was performed by taking all of the target genes predicted as part of an EPI in that cell type and the top 5 with the lowest *p*-values are shown. Dotplots are shown for all target genes in the cell type’s EPI network (**A**) and for only the target genes predicted to interact with cell-type-specific enhancers (**B**).

**Figure 4 ijms-25-09840-f004:**
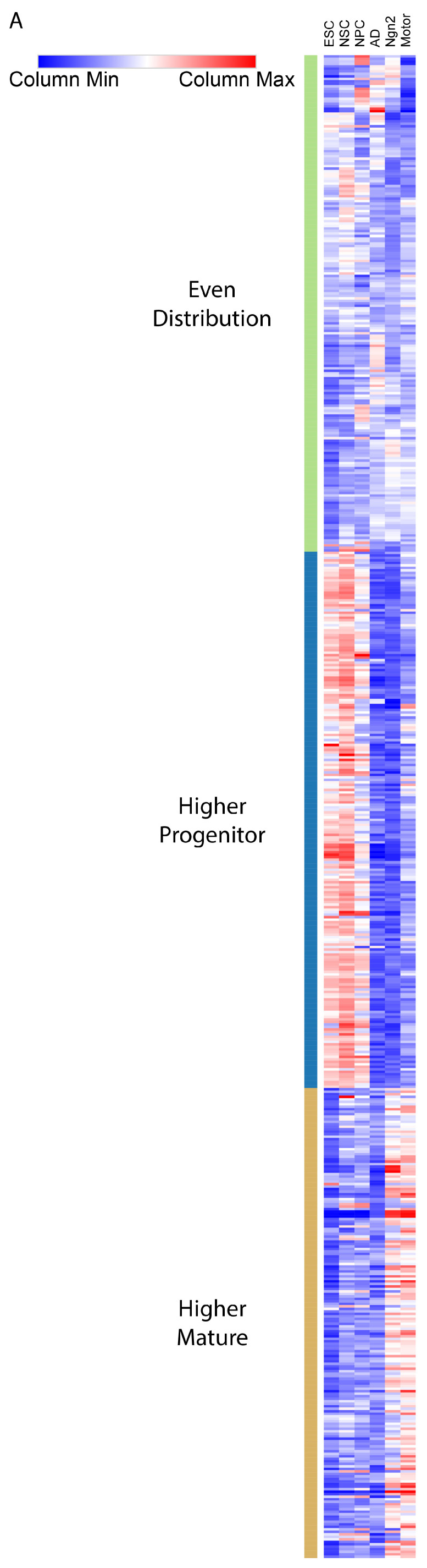
TF weighted binary matrix: (**A**) Heatmap displaying the amount of presence each TF (row) has within each cell-type group (column) From left to right the columns are ESC, NSC, NPC, AD, Ngn2, Motor. Values are the percentage of enhancers that the TF binds to that belong to that cell type, each column adding to 100%. Cell color denotes the relative percentage of TF binding within that cell type. A higher percentage of the TF binding uniquely to that cell type results in more red and less blue coloring. TF columns are organized via hierarchical clustering to identify clusters of TFs binding preferentially to certain cell types. (**B**) Gene Ontology Enrichment Dot plot for the top 10 enriched processes of the 2 TF subsets. The higher progenitor column consists of the TFs that have the highest relative percentage of predicted binding to enhancers within progenitor cell types and lower relative binding percentages in mature neurons (center section of Figure 4A). The higher mature column consists of the TFs that have the highest relative percentage of predicted binding to enhancers within mature neuron cell types and lower relative binding percentages in progenitor cell types (bottom section of Figure 4A).

**Figure 5 ijms-25-09840-f005:**
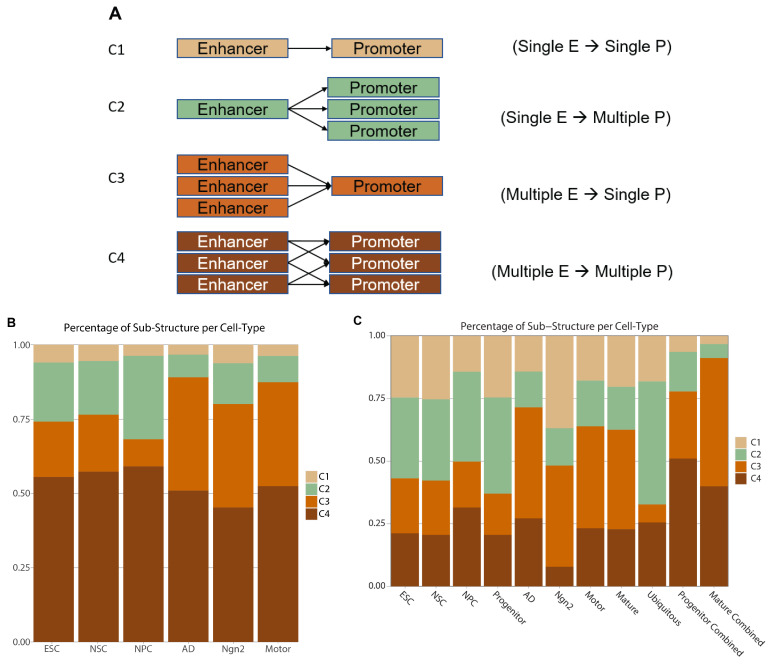
EPI sub-structure: Each EPI is assigned to a sub-structure group (**A**) based on the number of promoters with which the EPI enhancer interacts and the number of enhancers with which the EPI promoter interacts. (**B**) Stacked bar plot breaking down the number of EPIs per cell type in the collapsed EPI network that belong to each sub-structure group. (**C**) Stacked barplot breaking down the number of EPIs per cell type in the subset EPI network that belong to each sub-structure group. The groups Progenitor and Mature consist of EPIs predicted active in at least 2 of the 3 cell types belonging to the progenitor or mature differentiation stage group, respectively. The Ubiquitous group consists of EPIs predicted to be active in at least 5 of the 6 cell types. The Progenitor Combined group consists of the union of EPIs from the ESC, NSC, NPC, and Progenitor subsets. The Mature Combined consists of the union of EPIs from the AD, Ngn2, Motor, and Mature subsets. Progenitor Combined and Mature Combined are included to represent the overlapping nature of the collapsed network.

**Figure 6 ijms-25-09840-f006:**
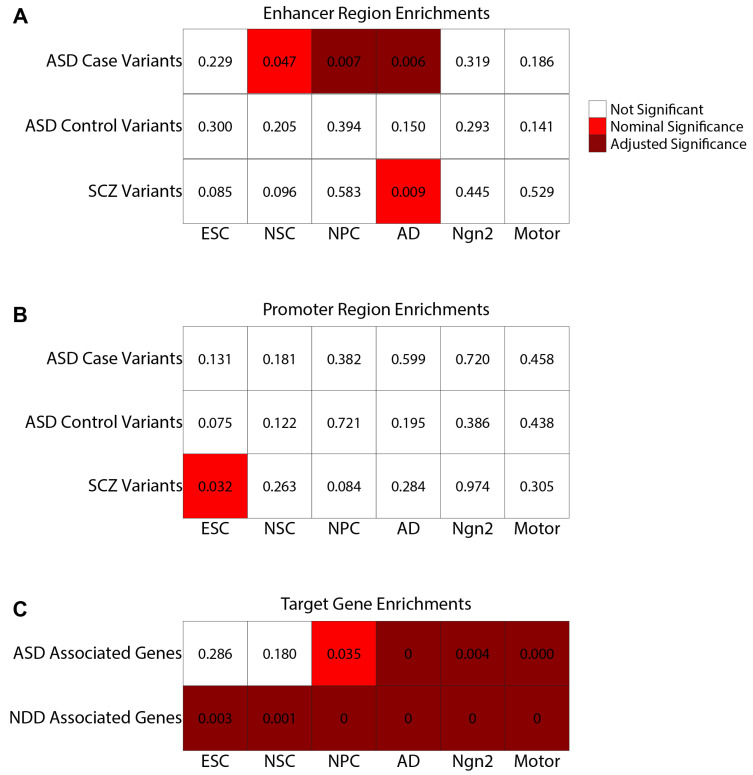
Empirical *p*-values of disease variant overlap. This heatmap denotes the empirical *p*-value found after 1000 repetitions of randomly selecting enhancer (**A**), promoter (**B**), or target gene (**C**) regulatory regions and identifying the number of regions that contain a variant (**A**,**B**) or have been previously associated with ASD or NDD (**C**). *p*-values calculated as 1 minus the number of iterations where the randomly selected regulatory regions contained equal or more variants than observed, divided by 1000. *p*-values less than or equal to the adjusted threshold of 0.008333 are shaded in dark red. *p*-values of nominal significance between 0.008333 and 0.05 are shaded in bright red. The ASD variant space consisted of ~250,000 variants, with an approximately even split between case and control variants. The Schizophrenia variant space consisted of ~30,000 variants from GWAS-identified variants along with variants within LD > 0.8 of the identified variants. The ASD-associated gene list consists of 252 genes. The NDD-associated gene list consists of 817 genes including the 252 ASD-associated genes.

**Figure 7 ijms-25-09840-f007:**
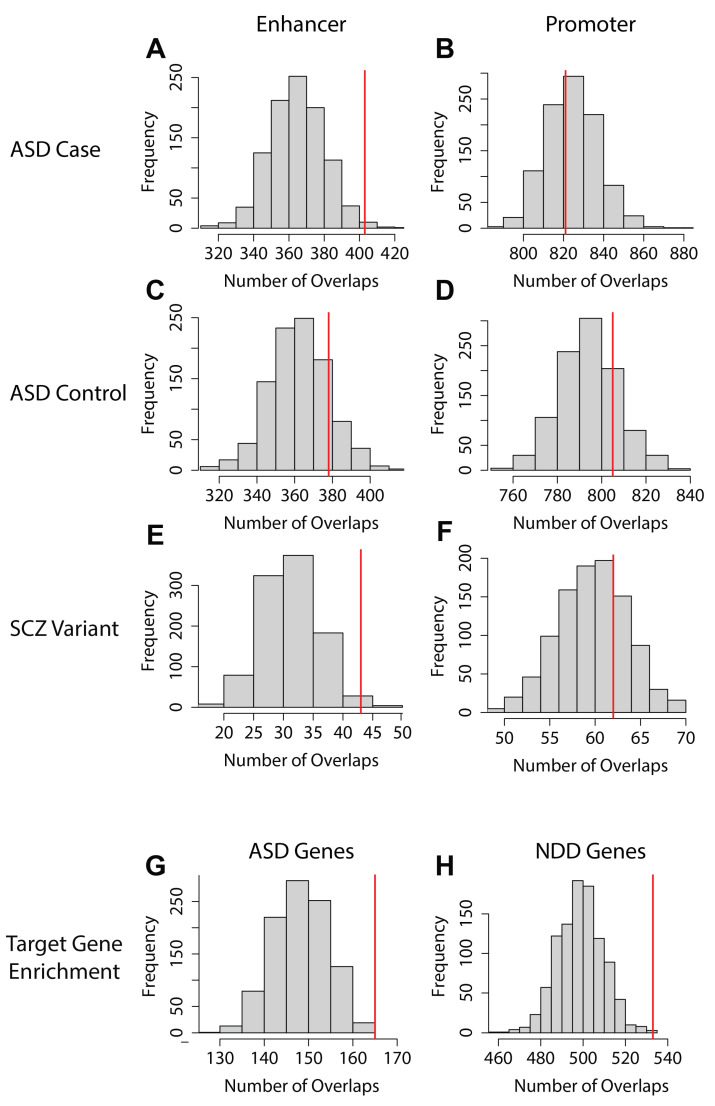
Representative empirical analysis histograms: We highlight the empirical analysis histograms of the AD cell type as a representative of the analysis performed on the 6 cell types. The AD cell type was selected due to it having a significant enrichment for ASD cases (**A**) and not ASD controls (**B**), nominal enrichment for SCZ variants in enhancers (**E**), and enrichment for ASD- and NDD-associated genes (**G**,**H**). The rows of the figure for panels A–F represent the variants being analyzed for enrichment with the columns being the regulatory regions that the variants might be enriched in. For panels G and H, they are both testing the target genes of the AD EPI network and the columns denote if it is being analyzed for enrichment of ASD-associated genes (**G**) or NDD-associated genes (**H**). The *X*-axis of the histograms represents the number of regions containing variants (**A**–**F**) or disease-associated genes (**G**,**H**). The *Y*-axis of the histograms represents the frequency of the 1000 randomly selected iteration observed values. The red line represents the observed number of enhancer (**A**,**C**,**E**), promoter (**B**,**D**,**F**), or target-gene (**G**,**H**) regions containing variants or disease-associated genes in the AD cell type’s EPI network.

**Table 1 ijms-25-09840-t001:** Description of cell types utilized in this study. This table includes the cell types included in this study. The table also includes the label of the cell type in the original publication and the accession number where the data was downloaded from.

Cell Type	Labeling in Original Publication	Data Source
Embryonic stem cells (ESC)	0 h	GSE115046
Neural stem cells (NSC)	72 h	
Neural Progenitor Cell (NPC)	N2	GSE110758
GABAergic Neurons	AD	GSE196856
Glutamatergic Neurons	Ngn2	
Motor Neurons	Motor	GSE113483

For clarity, throughout this paper, neural progenitor cells (NPC) are referred to as N2.

## Data Availability

The raw data collected and analyzed in this study are available in the Gene Expression Omnibus at https://ncbi.nlm.nih.gov/geo/ (accessed on 28 November 2022), under the accession numbers: GSE115046, GSE110758; GSE196856, and GSE113483.

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
