# Peer review of "Network Analysis of Enhancer–Promoter Interactions Highlights Cell-Type-Specific Mechanisms of Transcriptional Regulation Variation"

_ijms, 2024, doi:10.3390/ijms25189840_

Round 1
Reviewer 1 Report
Comments and Suggestions for Authors
· The abstract needs clearer conclusions. e.g. EPIs can identify disease-associated variants…etc. how these findings might be applied in practical settings, such as drug development or diagnostic tools.
· Introduction need section about challenges in studying enhancer and promoter (EP) interactions, especially in the context of disease.
· Introduction some parts Line 76-84 better belong to methods and similarly Line 85-95 could be used for conclusions.
· Methods need some details. First, the use of the Activity By Contact (ABC) model for predicting EPIs. Details needed for the thresholds for ABC score (< 0.02) and RNA expression (less than 1 TPM). Is these based on previous studies or specific to this dataset. Second, in the generation of EPI networks. Jaccard index need further details about overlap between EP. Also Jaccard index seems low? Would not be additional testing?
· Discussion can expand on limitations in using the ABC model. Suggest ideas for future research.
· Figures are too small.
· Data availability statement seems contain repetition.
Author Response
Network Analysis of Enhancer-Promoter Interactions Highlights Cell-Type Specific Mechanisms of Transcriptional Regulation Variation IJMS round 1 reviewer comments
Reviewer 1:
- The abstract needs clearer conclusions. e.g. EPIs can identify disease-associated variants…etc. how these findings might be applied in practical settings, such as drug development or diagnostic tools.
We thank the reviewer for their comments. We have added to the abstract about the practical applications of our findings on lines 37-41.
- Introduction need section about challenges in studying enhancer and promoter (EP) interactions, especially in the context of disease.
We appreciate the request to add more information about the challenges of investigating EP interactions. We mention on lines 49-55 the challenges of studying EP interactions, and have added the complexity of researching in a disease context to lines 60-62.
- Introduction some parts Line 76-84 better belong to methods and similarly Line 85-95 could be used for conclusions.
We thank the reviewer for the suggestion to move sections of the introduction to the methods and conclusions. While we do have statements similar to these in the methods and conclusions, we believe that having these present in the introduction allows for readers to get a helpful overview and understanding of the analysis and results prior to the detailed results and conclusions sections.
- Methods need some details. First, the use of the Activity By Contact (ABC) model for predicting EPIs. Details needed for the thresholds for ABC score (< 0.02) and RNA expression (less than 1 TPM). Is these based on previous studies or specific to this dataset. Second, in the generation of EPI networks. Jaccard index need further details about overlap between EP. Also Jaccard index seems low? Would not be additional testing?
We thank the reviewer for the request to clarify the filtering criteria. We have added statements clarifying out choice of an interaction score > 0.02 and RNA TPM > 1 to lines 547-553.
Regarding the Jaccard index, the low values represent a low overlap of enhancers or promoters overlapping between two given cell types. We highlight that this finding is consistent in our upset plots (supplemental figures S2) which finds that there is a low number of interactions between differing number of cell types which would lead to a low Jaccard index value. We also find consistent results with our subset EPI network (supplemental figure S3), in which we remove duplicate EPIs between cell types, to reduce noise in the amount of enhancers and promoters that overlap between cell types. We added a statement on lines 198-200 to highlight this finding.
- Discussion can expand on limitations in using the ABC model. Suggest ideas for future research.
We thank the reviewer for their suggestion and have added limitations and potential future analyses to the discussion on lines 437-448.
- Figures are too small.
We thank the reviewer for bringing this to our attention. We have increased the label size for figures 2A and 2B as well as adding information to the figure legend of 4A on lines 268-269. We have rearranged some of the figures and made them larger but the final decision about figure size and placement may be up to the editor.
- Data availability statement seems contain repetition.
We thank the reviewer for highlighting this issue. We have update the statement on lines 706-708 to be more concise.
Reviewer 2 Report
Comments and Suggestions for Authors
In this original article entitle “Network Analysis of Enhancer-Promoter Interactions Highlights Cell-Type Specific Mechanisms of Transcriptional Regulation Variation”, Koesterich et al. studied the interaction of enhancers and promoters within six different neuronal cell types spanning different neuronal differentiation stages. The authors presented a detailed methodology to predict Enhancer-Promoter Interactions that can be useful to prioritize Single Nucleotide Polymorphisms associated with diseases and further understand the enhancer-promoter dynamics during cell type specification and how this may be disrupted during disease.
In summary, they found interesting results showing that enhancer-promoter interactions are highly associated with cell type specificity. Moreover, enhancers rather than promoters or transcription factors highly contribute to cell type specificity. Enhancers may be regulating a particular set of promoters and have different transcription binding sites that together with the gene expression of transcription factors will contribute to cell type specificity. Furthermore, the authors show that each part of the predicted EPIs: enhancers, promoter regions, and target genes, can be utilized as a selection mechanism for disease associated variants and genes, opening the possibility to improve selection and prioritization of variants for testing using functional assays. In addition to these interesting findings, the Methods section is nicely explained including important details for processing the data used in the study.
Nonetheless there are some minor issues that I would like to report:
1. Line 85-87: This phrase is difficult to understand. I suggest changing it to: “Our results suggest that enhancers, rather than promoters and TFs computationally predicted to bind enhancers, are the regulatory elements showing higher cell type specificity.”
2. Line 185-188: It is not clear in which Figure we can observe this increase of unique enhancers. I suggest adding an additional reference to Supplementary Figure 3A: “Within this subset network analysis, we observed an increased amount of unique enhancers within each of the nine cell type categories (Supplemental Figure 3A) compared to the findings of the collapsed network EPI categories (Figure 2B), while the proportion of unique promoters remains largely unchanged (Supplemental Figure 3A).”
However, in order to reduce the word number, it is also suitable to put both references together at the end of the paragraph: “Within this subset network analysis, we observed an increased amount of unique enhancers within each of the nine cell type categories compared to the findings of the collapsed network EPI categories, while the proportion of unique promoters remains largely unchanged (Supplemental Figure 3A, Figure 2B).”
3. Line 214: The space between “Figure 3.” and “Dotplot” is underlined.
4. Line 481-482: Instead of a DOI link I would include the reference to the paper: “We then removed duplicated data and reads in the ENCODE blacklisted regions [REF] with GATK [39].”
5. Finally, there is no code availability. Despite it is not mandatory, I consider it can be a useful tool for the scientific community for performing similar analyses on other cell types an diseases.
Taking all into account, I considered that this manuscript can be a good contribution for the special issue “New Mechanisms and Therapeutics in Neurological Diseases 3.0” of the International Journal of Molecular Sciences and should be considered for publication.
Author Response
Network Analysis of Enhancer-Promoter Interactions Highlights Cell-Type Specific Mechanisms of Transcriptional Regulation Variation IJMS round 1 reviewer comments
Reviewer 2:
- Line 85-87: This phrase is difficult to understand. I suggest changing it to: “Our results suggest that enhancers, rather than promoters and TFs computationally predicted to bind enhancers, are the regulatory elements showing higher cell type specificity.”
We thank the reviewer for their suggestion and have made the corresponding edits to the statement now on line 94-96.
- Line 185-188: It is not clear in which Figure we can observe this increase of unique enhancers. I suggest adding an additional reference to Supplementary Figure 3A: “Within this subset network analysis, we observed an increased amount of unique enhancers within each of the nine cell type categories (Supplemental Figure 3A) compared to the findings of the collapsed network EPI categories (Figure 2B), while the proportion of unique promoters remains largely unchanged (Supplemental Figure 3A).”
However, in order to reduce the word number, it is also suitable to put both references together at the end of the paragraph: “Within this subset network analysis, we observed an increased amount of unique enhancers within each of the nine cell type categories compared to the findings of the collapsed network EPI categories, while the proportion of unique promoters remains largely unchanged (Supplemental Figure 3A, Figure 2B).”
We thank the reviewer for requesting this to be more clearly labelled. We have added the supplemental Figure 3A to the line now on lines 195-196.
- Line 214: The space between “Figure 3.” and “Dotplot” is underlined.
We thank the reviewer for finding that and have made the correction.
- Line 481-482: Instead of a DOI link I would include the reference to the paper: “We then removed duplicated data and reads in the ENCODE blacklisted regions [REF] with GATK [39].”
We thank the reviewer for the suggestion and have changed the link to a reference and updated the citation numbers accordingly.
- Finally, there is no code availability. Despite it is not mandatory, I consider it can be a useful tool for the scientific community for performing similar analyses on other cell types an diseases.
We thank the reviewer for the suggestion and agree that it can be a useful tool to the science community. We are currently in the process of formatting and uploading the codes used in this study and have added a code availability statement with the location of these uploaded files on lines 712-713.